

# Limited influence of marine sediment lyophilization on prokaryotic community structure assessed via amplicon sequencing: an example from environmentally contrasted sediment layers in Toulon harbor (France)

Benjamin Misson[1], Cédric Garnier[1] and Alexandre J. Poulain[2]

[1] Université de Toulon, Aix Marseille University, CNRS, IRD, MIO, Toulon, France
[2] Department of Biology, University of Ottawa, Ottawa, ON, Canada

## ABSTRACT

Sediment lyophilization is a common process that allows for long-term conservation and sharing of marine sediments for multiple downstream analyses. Although it is often used for geochemical studies, the effects of lyophilization on prokaryotic taxonomic diversity assessment remained to be assessed. Here, we tested the effect of lyophilization on microbial diversity assessment using three sediment layers corresponding to various sediment ages and chemical contamination levels sampled from a marine Mediterranean harbor. Duplicate DNA samples were extracted from wet frozen or lyophilized sediments, and 16S rRNA gene amplicon sequence variants were analyzed. We detected changes in community structure over depth linked to both dominant and less abundant taxa whether sediments were lyophilized or not. Data from both wet frozen and lyophilized sediments led us to conclude that historical chemical contamination of the sediment of Toulon Bay did not appear to be the main environmental variable shaping prokaryotic community structure on the vertical dimension, but that sediment diagenesis was. We conclude that sediment lyophilization is compatible with marine biogeochemical and ecotoxicological studies but that caution should be used when discussing small variations among samples.

## INTRODUCTION

Linking chemical contamination to ecological effects through genomic approaches remains challenging. Multidisciplinary approaches are needed, but their application often requires that varying sample amounts and sometimes different conservations means be used. Marine sediment conservation for inorganic contamination evaluation often require freeze-drying (*Tessier et al., 2011*; *Dang et al., 2015*; *Outridge et al., 2017*). In order to couple microbial community structure analyses with geochemical investigations, the same

Corresponding author
Benjamin Misson,
misson@univ-tln.fr

samples need to be processed for different downstream analyses. One question that microbiologists interested in geochemical processes often face is whether sediment freeze-drying, typically required to determine metal burden per dry mass of sediment, is compatible with microbial studies. If yes, such sample conservation could facilitate collaborative work performed by distant laboratories.

A previous study on freeze-dried soil and freshwater sediment samples (*Miller et al., 1999*) demonstrated the ability to use freeze-dried samples to extract and amplify microbial DNA, but no comparison with results obtained with wet samples was provided. More recently, *Gianaroli et al. (2012)* demonstrated by using microscopy that DNA integrity was not altered in freeze-dried eukaryotic cells when compared to fresh ones. With recent advances in high throughput sequencing and downstream bioinformatics analyses, we can question whether more subtle consequences on DNA structure or extraction could interfere with microbial diversity assessments because of sediment freeze-drying.

A recent study targeting soil samples, using 454 pyrosequencing and focusing on nucleic acids conservation, demonstrated that alpha- and beta-diversity were little affected by freeze-drying (*Weißbecker, Buscot & Wubet, 2017*). However, the taxa encountered were not reported. Given the very large microbial diversity in soil or sediment, it is reasonable to expect differential conservation of microbial DNA upon freeze-drying and/or recovery during DNA extraction based on cell wall properties or even DNA sequences (e.g., low vs. high GC).

Furthermore, marine coastal sediments are rich in salts and humic acids, potentially inhibiting PCR amplification (*Kreader, 1996*), and increasingly subject to anthropogenic contaminants, among which divalent metallic cations can negatively alter nucleic acids recovery during the extraction procedure (*Stein et al., 2001*). These chemicals are further concentrated during sediment freeze-drying, thus potentially challenging microbial community structure assessment. Validating the suitability of molecular tools on DNA extracted from freeze-dried sediments is still necessary (*Bey et al., 2010*).

This study aimed at comparing prokaryotic community diversity determined by MiSeq high throughput sequencing of 16S rRNA gene amplicon from DNA extracted from wet frozen or freeze-dried marine sediments. Our analysis focused on alpha and beta diversity metrics. We used highly contaminated sediments from the Toulon Harbour (NW Mediterranean Sea, France) which anthropogenic recent and historical chemical contamination was previously characterized (*Misson et al., 2016*). In addition to testing for the role of lyophilization on diversity assessment, we were interested in evaluating the relative contribution of this contamination and of naturally occurring processes such as diagenesis on prokaryotic diversity. To do so, we used three sediment layers spanning 150 years of sedimentation and capturing the main contamination event associated with the scuttling of the French fleet during World War II.

## MATERIALS AND METHODS

### Sediment sampling and conditioning

The sampling site is located in the heavily multi-contaminated Toulon Bay (France), at sampling station #12, (43°6′34.9″N, 5°55′41.1″E) previously intensively studied by *Dang*

*et al. (2015)*. A 50 cm-long and 10 cm-diameter sediment core was sampled in a PlexiGlass® tube with the help of divers in September 2015. Additional details concerning the chemical contamination in this area of Toulon bay can also be retrieved from other studies (*Tessier et al., 2011*; *Pougnet et al., 2014*; *Misson et al., 2016*; *Wafo et al., 2016*). Sampling was performed with the authorization of the Laboratoire d'Analyse, de Surveillance et d'Expertise de la Marine (LASEM) Nationale de Toulon.

The core was sliced under nitrogen-enriched atmosphere using sectioning material previously sterilized with bleach and cleaned for trace metal sampling with trace metal grade nitric acid. The core was sectioned every 1 cm from depths 0 to 20 cm and subsequently every 2 cm from depths 20 to 46 cm. Half of each sediment slice was immediately stored at −20 °C and freeze-dried for a period of 72 h under a vacuum of $10^{-5}$ atm with a Pilot PCCPLS15 freeze dryer (Cryotec) and then homogenized. These lyophilized sediments were further used for Hg content determination as well as DNA extraction. The other half (subsequently called "frozen" sediments) was immediately rinsed with a sterile buffer (SB) composed of 10 mM EDTA, 50 mM TrisHCl and 50 mM $Na_2HPO_4$ at pH 8 to remove contaminants such as humic substances and divalent cations (*Brazeau et al., 2013*). After centrifugation at $8,000 \times g$, the pellet was stored at −20 °C until DNA extraction.

## Hg analyses

Total Hg in sediments was analyzed as previously described (*Poulain et al., 2015*). Specifically, total mercury was analyzed by thermal decomposition with gold trap amalgamation and cold vapor atomic absorption method (UOP Method 938-00, detection limit of 0.01 ng Hg and range up to 1,000 ng Hg) using a Nippon Instruments Corporation's Mercury SP-3D Analyser (CV-AAS). The instrument was calibrated with Mercury Reference Solution 1,000 ppm ± 1% (Fisher CSM114-100) and MESS-3 ($91 \pm 9$ ng $g^{-1}$, National Research Council of Canada) was used as reference material. Blanks were performed as suggested by the manufacturer.

## DNA extraction

To compare samples from different ages and contamination levels, three sediment slices were chosen: superficial sediments corresponding to recent deposits (0–1 cm, <5 years according to settling rates in the bay; *Tessier et al., 2011*), mid-depth sediments corresponding to the historical contamination event (10–11 cm, see the results section, ~75 years old) and deep sediments corresponding to an older period with a limited human impact on the site (19–20 cm, ~150 years old; Fig. 1).

For each sediment slice, two subsamples were taken from lyophilized material and two others from frozen material, leading to a total of 12 samples treated for molecular analyses. DNA was extracted from 250 mg of freeze-dried or 250 mg of frozen sediments. After weighting, freeze-dried sediments were rehydrated in 400 μL of SB for 1 h at 4 °C. Then DNA extraction was performed concomitantly by two experimentalists using PowerSoil DNA Isolation Kit (MoBio) according to manufacturer's protocol. DNA

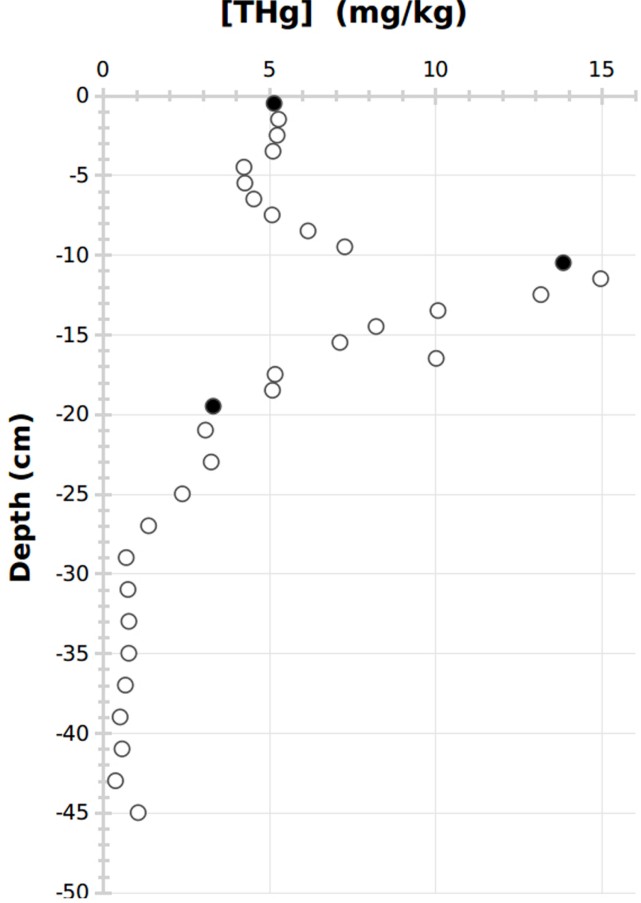

**Figure 1 Vertical variations of total Hg content in the sediment core.** Closed circles represent the sediment slices used for DNA sequencing.

concentration and purity were estimated using a NanoDrop ND 1,000. DNA extracts were then stored at −20 °C until further use.

## Quantification of bacterial 16S rRNA gene copies

To evaluate whether sediment lyophilization could have reduced the efficiency of DNA extraction, bacterial 16S rRNA gene copies were quantified using qPCR and primers BAC338f/515R (*Borrel et al., 2012*). Amplification reactions were performed in triplicate in a LightCycler 480 thermocycler (Qiagen) with GoTaq SybrGreen mastermix (Promega, Madison, WI, USA) in a final volume of 10 μL containing 0.25 μM of each primer and two μL of 20-fold diluted DNA extract to avoid PCR inhibition. Serial 10-fold dilutions of a linearized recombinant plasmid ranging from $10^7$ to $10^2$ copies were also amplified in triplicate to produce a standard curve used for determining 16S rRNA gene copy number in the samples. According to the standard curve slope, the reaction yielded an efficiency of 97%.

16S rRNA gene copy numbers were reported to the mass of frozen sediments used for extraction. In the case of lyophilized samples, the water content of the sediment (64%)

observed previously at the same site was used to calculate the equivalence between dry and wet weights (*Tessier et al., 2011*).

## Evaluation of prokaryotic community diversity

Amplification of the V4–V5 region of 16S rRNA gene targeting bacteria and archaea was performed using primers 515F-Y/926R (*Parada, Needham & Fuhrman, 2016*). Reaction mixtures contained 30–60 ng of DNA, 2× GoTaq Long PCR Master Mix (Promega, Madison, WI, USA) and 0.4 μM of each primer, in a final volume of 60 μL. The PCR program included an initial heating step of 2′ at 95 °C followed by 25 cycles of 95 °C for 30″, 50 °C for 45″ and 72 °C for 45″, and a final extension of 10′ at 72 °C. PCR amplification efficiency and specificity was checked after migration of five μL of PCR products on a 1.5% agarose gel. Extraction blank performed with MilliQ water as well as no-template PCR control did not provide any amplification products. Amplicons were then purified, fuzed with Illumina barcodes and paired-end sequenced (2 × 250 bp) with the chemistry V2 and an Illumina MiSeq sequencer by the GeT-PLaGe platform (Castanet-Tolosan, France).

MiSeq raw reads were analyzed with DADA2 (*Callahan et al., 2016*) in RStudio (*R Core Team, 2017*). Sequences were trimmed according to average Q-scores, then filtered allowing no N, no more than 2 expected errors whatever the read (i.e., forward or reverse) and a minimal Q-score of 3. Taxonomic information was assigned to ASVs with the SILVA v.132 database (*Pruesse et al., 2007*; *Quast et al., 2013*). Sequences classified as mitochondria or chloroplasts were removed from the ASV table. Subsampling at the same sequencing depth for all sample did not appear necessary for further alpha and beta diversity computation, because rarefaction curves of each sample showed a plateau (see the "Results" section). Sequencing reads were deposited in the National Center for Biotechnology Information Sequence Read Archive (NCBI SRA) under the accession number PRJNA599066. Rarefaction curves, alpha diversity calculations (including Shannon and reciprocal Simpson's indexes), weighted UniFrac distances and PCoA were calculated and constructed with RStudio using the *phyloseq* (*McMurdie & Holmes, 2013*) and *vegan* (*Oksanen et al., 2019*) packages.

# RESULTS

## Historical contamination profile

Concentrations of HgT spanned 2 orders of magnitude, ranging from 0.37 to 15 ppm. The vertical profile exhibited stable and low HgT concentrations (0.4–1.1 ppm) at the bottom of the core (from −46 to −28 cm), then a first increase to 3.3 ppm was recorded (from −28 to −19 cm) before a second but sharper increase towards the maximal historical contamination (recorded at −12 cm). Concentrations subsequently decreased from ca. 12 cm to stabilize at ~5 ppm in the top 8 cm (Fig. 1).

## Prokaryotic community abundance and diversity

Bacterial 16S rRNA gene copy number was recorded as a proxy of prokaryotic abundance. Once normalized to sediment wet weight (or equivalent) used for extraction, values ranged

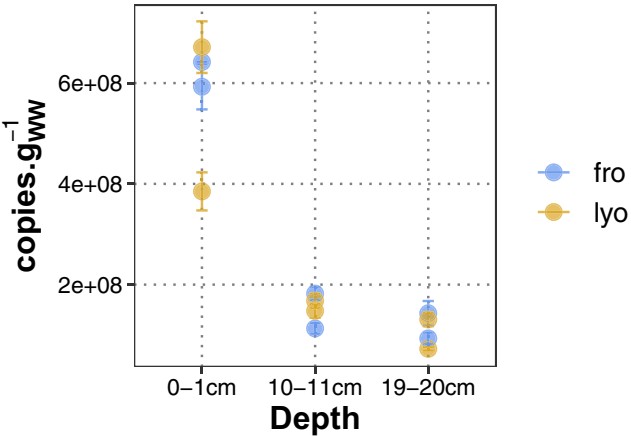

**Figure 2 Bacterial 16S rRNA gene abundance.** "lyo" refers to lyophilized sediments, "fro" refers to frozen sediments.

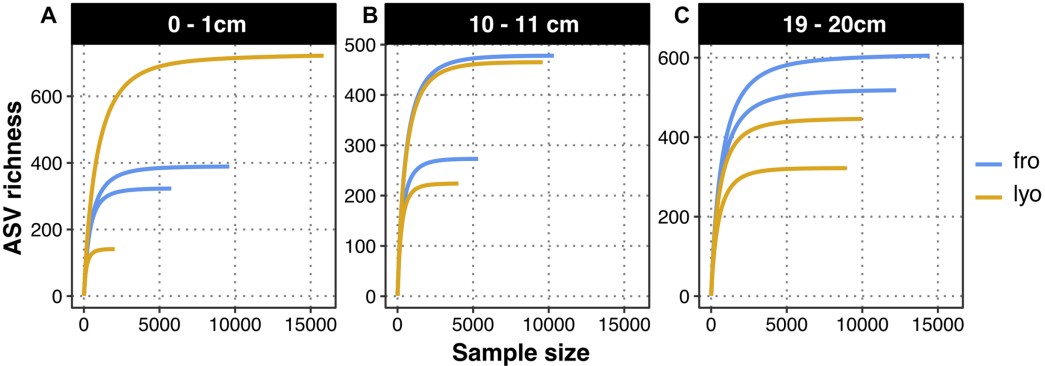

**Figure 3 Comparison of rarefaction curves according to sediment treatment and depth.** (A) Surficial sediment slice (0–1 cm). (B) Intermediate sediment slice (10–11 cm). (C) Deep sediment slice (19–20 cm). "lyo" refers to lyophilized sediments, "fro" refers to frozen sediments.

from 0.93 to $6.42 \times 10^8$ copies $g^{-1}$ and from 0.73 to $6.71 \times 10^8$ copies $g^{-1}$ for frozen and lyophilized sediments, respectively. Whatever the sediment conditioning before extraction, a similar decreasing trend with depth was observed, especially between 0–1 cm and 10–11 cm (Fig. 2). In spite of this common trend, a higher variability was recorded for lyophilized sediment from 0 to 1 cm.

Sequencing the prokaryotic community yielded 2,396 ASVs from 108,216 sequences. Rarefaction curves demonstrated that a plateau was reached quickly for all samples (Figs. 3A–3C). The observed richness per sample ranged between 141 and 722 ASVs. As observed for 16S rRNA gene copies, the observed richness at 0–1 cm after lyophilization appeared highly variable (Fig. 4A). However, it should be noted that the sample showing the lowest richness was not the one showing the lowest 16S rRNA gene copy number. At 10–11cm, both observed richness values and variability appeared similar for frozen and lyophilized sediment (Fig. 4B). At 19–20 cm, the variability among replicates was similar for both frozen and lyophilized sediment, but the observed richness appeared

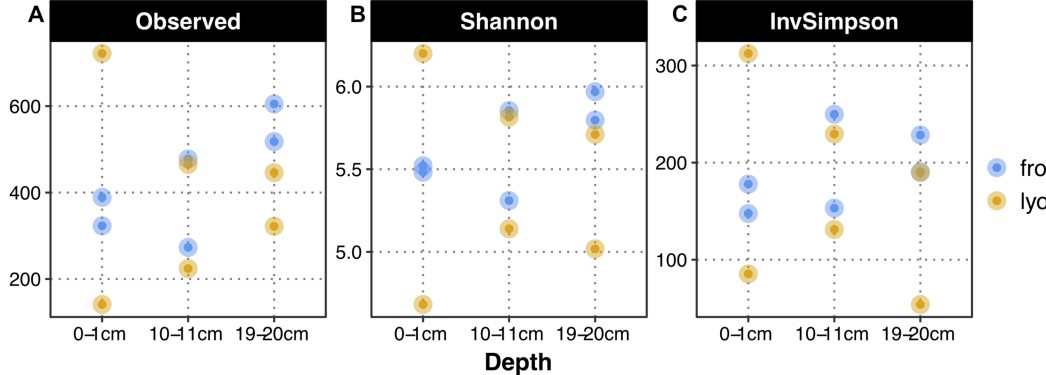

**Figure 4 Alpha diversity metrics as a function of sediment treatment and depth.** (A) Observed ASV richness. (B) Shannon diversity index. (C) Reverse Simpson diversity index. "lyo" refers to lyophilized sediments, "fro" refers to frozen sediments.

higher for lyophilized sediments. Other alpha diversity metrics, Shannon and reciprocal Simpson indexes, presented a higher variability for lyophilized than for frozen sediment at 0–1 cm, and also at 19–20 cm. We observed far lower values of Shannon and reciprocal Simpson indexes for one replicate at 19–20 cm (Fig. 4C). Finally, DNA extracted from frozen sediment led to a slight increasing trend with depth for all metrics, a trend that could not be observed from lyophilized material because of the higher variability observed.

ASVs were distributed among 49 phyla corresponding mainly to *Bacteria*. Indeed, after extraction from frozen sediment, the proportion of archaeal sequences ranged from 0.3% to 14.0%. Archaeal ASVs were more represented at 10–11 cm and 19–20 cm (9.4–14.0%) than in the surface layer (0.3%). The same trend was observed for lyophilized sediments, *Archaea* being undetected or rare at 0–1 cm (0–0.3%), and more abundant in intermediate (12.3–14.3%) and deep layers (13.5–17.4%). Among the bacterial phyla encountered, *Proteobacteria* (especially *Delta-* and *Gammaproteobacteria*), *Chloroflexi* (especially *Dehalococcoidia*) and *Bacteroidetes* were dominant and accounted for 40.8–70.9% of the community recovered from frozen sediment (Fig. 5). Strong vertical variations in community structure were recorded, mainly between the surface layer, on the one hand, and intermediate and deep layers, on the other. *Gammaproteobacteria* and *Bacteroidetes* were dominant groups in the surface layer (24.1–38.5% and 12.7–13.2%, respectively) but largely decreased in intermediate and deep layers (0.2–10.8% and 1.5–2.6%, respectively). We noted that *Dehalococcoidia* strongly increased in intermediate and deep layers (16.6–18.9% and 23.6–23.8%, respectively) when compared to the surface (from undetected to 0.3%). This vertical structure shift appeared highly reproducible between frozen and lyophilized sediment samples at all depths (Figs. 5A–5C), although the proportion of *Gammaproteobacteria* in one deep lyophilized sample appeared abnormally high (12.1%). In the same sample, *Firmicutes* reached 16.7% while they were not detected in the other samples from the same depth.

Taking into account the whole community and not only dominant phyla or classes, weighted UniFrac distance analysis performed at the ASV level largely confirmed both the vertical trend, and the similarities observed between lyophilized and frozen sediments.

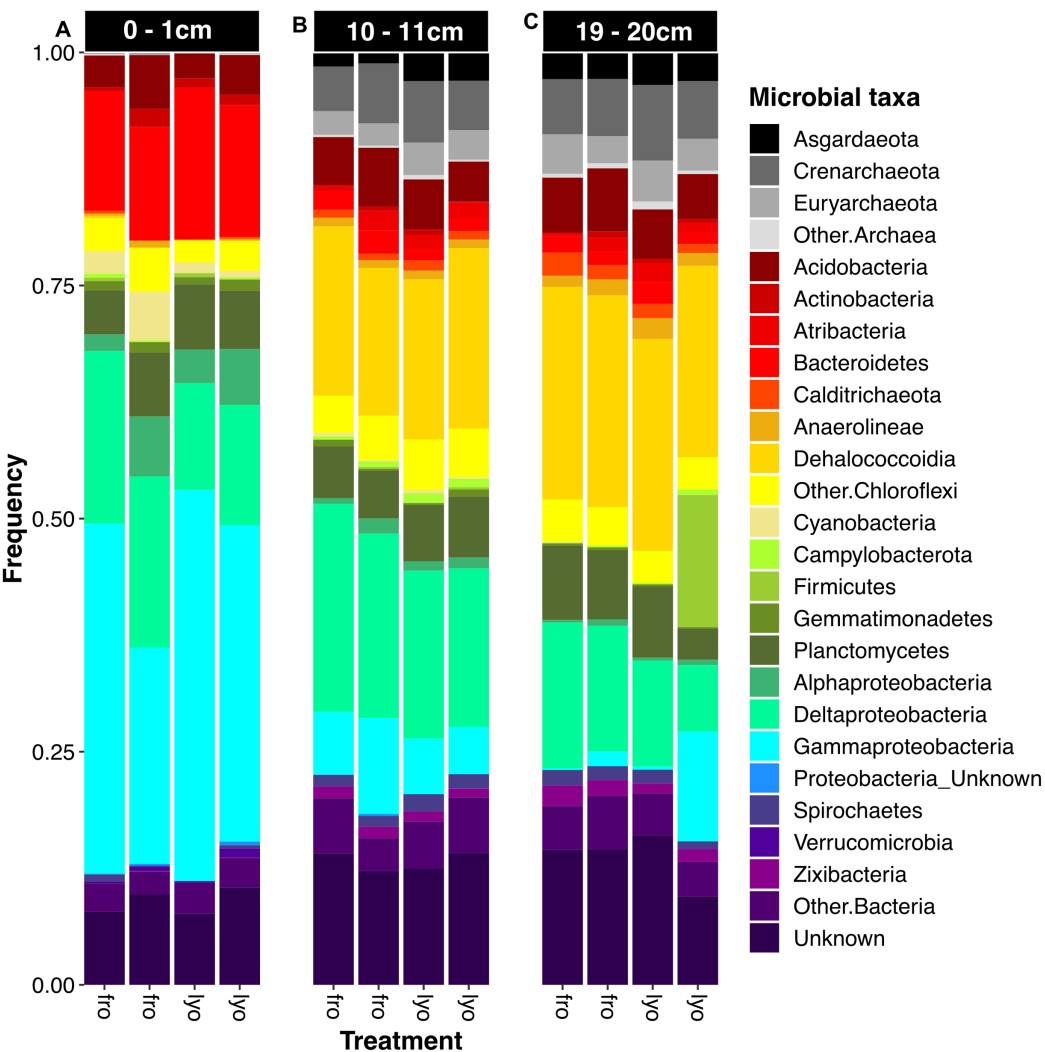

**Figure 5 Community structure with phylum or class contribution according to sediment treatment and depth.** (A) Surficial sediment slice (0–1 cm). (B) Intermediate sediment slice (10–11 cm). (C) Deep sediment slice (19–20 cm). Duplicates were not grouped in order to allow the evaluation of the reproducibility of the analysis. Phyla or classes never representing more than 1% in any sample are grouped as "Other" for their corresponding domain or phylum. On the *x*-axis, "lyo" refers to lyophilized samples, "fro" refers to frozen samples.                

As demonstrated by Fig. 6, the large majority of community structure variation was recorded between the surface layer and the other two samples, their separation being evident on the first axis of the PCoA representing more than 90% of the variability whatever treatment before DNA extraction was applied (Figs. 6A and 6B). Some variability among replicates (at 0–1 cm for frozen sediments and 19–20 cm for lyophilized ones) could be observed along the second axis, representing 3.0–4.3% of the variability. When considering both treatments together (Fig. 6C), the second axis allowed to discriminate intermediate (10–11 cm) and deep samples (19–20 cm) as well as support the influence of sediment treatment (fro vs. lyo). Although this observation could suggest some phylogeny-related susceptibility to lyophilization, the small representation of
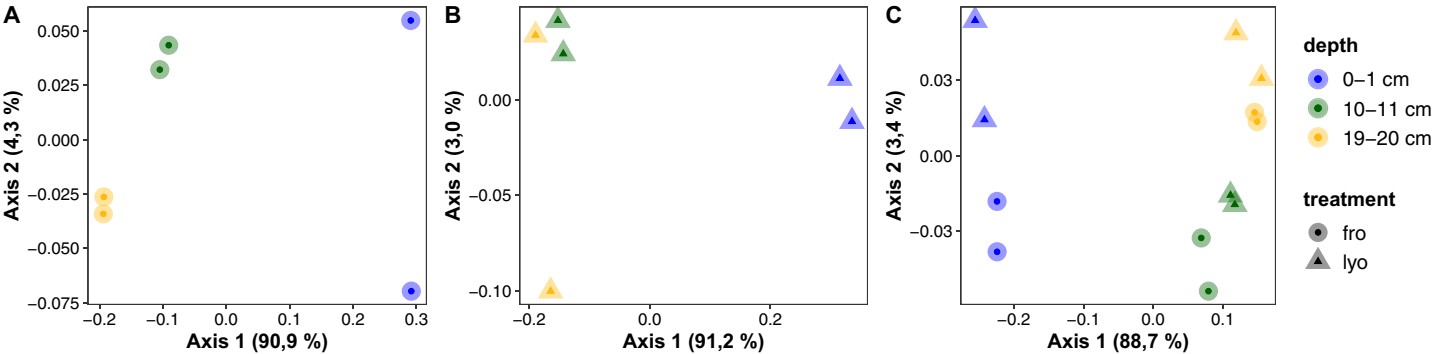

**Figure 6** PCoA representation of weighted UniFrac distance between samples according to sediment treatment or depth. (A) Frozen samples. (B) Lyophilized samples. (C) Both treatments together. "lyo" refers to lyophilized sediments, "fro" refers to frozen sediments.

the biological variability on the second axis (3.4%) tends to indicate that such difference could be restricted to rare groups and only slightly alter our assessment of community structure.

## DISCUSSION

In this study, we evaluated the influence of sediment freeze-drying on 16S rRNA gene amplicon diversity analyses using three sediment layers collected from a sediment core.

The highest congruence between results obtained from frozen and lyophilized sediments was for beta diversity analyses. Indeed, at the ASV level, within layer differences observed for frozen or lyophilized materials appeared lower than when comparing frozen vs. lyophilized samples. Very few differences were observed for dominant groups although some replicate showed singular community structure for either frozen or lyophilized sediment. We did not observe an effect of DNA extraction or degradation leading to a qualitative bias, which is in agreement with the absence of an effect of lyophilization on DNA structure within cells (*Gianaroli et al., 2012*).

When considering alpha diversity, sediment lyophilization resulted in more variability between replicates at the surface than for the two deeper layer samples. Moreover, vertical trends were not reproducible between frozen and lyophilized sediments. It should be noted that the number of ASVs retrieved from both lyophilized and frozen sediments (a few hundreds per samples) is in line with what should be expected and supports the suitability of DADA2, compared to prior programs that inflated diversity 10–100 times (*Edgar, 2013*).

Finally, qPCR analysis did not detect any major or recurrent modification of DNA extraction yield after sediment lyophilization. This suggests that differences in alpha or beta diversity could rather be linked to either natural variability, or to variability introduced by amplification and/or sequencing.

Together, these data suggest that lyophilization of marine sediments does not seem to lead to large biases in microbial taxonomic diversity evaluation. However, caution should be taken, especially when interpreting small amplitude variations of alpha diversity.

Having established that frozen and freeze-dried sediments were both amenable for 16S rRNA gene sequencing analyses, we studied the effect of a historical contamination gradient recorded vertically in Toulon Bay. Using $^{210}$Pb dating, *Tessier et al. (2011)* showed that Toulon Bay experienced an important chemical contamination event during World War II, most probably associated with the scuttling of the French fleet. It resulted in a strong metal enrichment of the sediment, particularly for total mercury (HgT) concentrations. As a consequence, (HgT) vertical profile has been a reliable dating tool for every new cores at this site, in addition to providing information on the contamination level of the samples. Our sediment sampling approach allowed us to compare prokaryotic taxonomic diversity in sediment settled before, during, or after this historical contamination event. In addition to the historical contamination gradient, sediment underwent diagenesis. Sediment diagenesis can lead to the development of a very reductive environment. According to a previous temporal survey performed at the same sampling site in Toulon Bay, sediment layers deposited before and during the historical contamination event are both anoxic and dominated by sulfate reduction. Closer to the surface, the sediment layer deposited after the contamination event corresponds to either oxic or suboxic conditions and is typically dominated by iron and manganese reduction due to its interaction with the oxic water column (*Dang et al., 2015*). These varying and contrasted geochemical conditions offered ideal conditions to evaluate how prokaryotic community structure changed over time and responded to chemical multicontamination. Indeed, both diagenesis and the varying toxicity of metal burden, this latter possibly attenuated by the presence of sulfides (*Dang et al., 2015*) and modulated by long term adaptation to metal stress (>70 years; *Misson et al., 2016*), could contribute to structuring the community.

We observed an important community structure shift between the top sediment layer on the one hand, and both intermediate and deep sediment layers on the other. We suspect that the strong diagenetic redox gradient induced by the very first stages of the early diagenesis (i.e., microbial mineralization of organic matter) is a very strong driver of prokaryotic community structure in the sediments of Toulon Bay as already demonstrated in various freshwater and marine sediments (*Cornall et al., 2013*; *Ruuskanen et al., 2018*; *Jiménez et al., 2018*). Subsequently, the distance analysis showed very slight discrimination of samples originating from before or contemporary to the historical contamination peak. These results suggest that the strong and multiple historical contamination of Toulon bay likely affected the prokaryotic benthic community, but far less than sediment aging and especially diagenesis-related redox gradients.

## ACKNOWLEDGEMENTS

We are grateful to the LASEM Toulon and French Navy divers for their help in sediment sampling, and to Philip Pelletier and Emmanuel Yumvihoze for their help with the analytical work. This work was performed in collaboration with the GeT core facility, Toulouse, France. We are indebted to the late Dr. Cedric Garnier, from whom we learned so much, and who thoroughly documented the sediment contamination of Toulon Bay, and made this study possible.

### Funding

This work was supported by a CARTT grant from the Institut Universitaire de Technology to Benjamin Misson and by an NSERC discovery and accelerator supplement grants to Alexandre J. Poulain. The funders had no role in study design, data collection and analysis, decision to publish, or preparation of the manuscript.

### Grant Disclosures

The following grant information was disclosed by the authors:
Institut Universitaire de Technology.
NSERC.

### Competing Interests

Alexandre Poulain is an Academic Editor for PeerJ

### Author Contributions

- Benjamin Misson conceived and designed the experiments, performed the experiments, analyzed the data, prepared figures and/or tables, authored or reviewed drafts of the paper, and approved the final draft.
- Cédric Garnier conceived and designed the experiments, performed the experiments, prepared figures and/or tables, and approved the final draft.
- Alexandre J. Poulain conceived and designed the experiments, performed the experiments, analyzed the data, prepared figures and/or tables, authored or reviewed drafts of the paper, and approved the final draft.

### Field Study Permissions

The following information was supplied relating to field study approvals (i.e., approving body and any reference numbers):

Sampling was performed with the authorization of the Laboratoire d'Analyse, de Surveillance et d'Expertise de la Marine (LASEM) Nationale de Toulon.

### DNA Deposition

The following information was supplied regarding the deposition of DNA sequences:

Sequencing reads are available at the National Center for Biotechnology Information Sequence Read Archive (NCBI SRA): PRJNA599066.

### Data Availability

Raw data are available in the Supplemental Files.

### Supplemental Information

Supplemental information for this article can be found online at http://dx.doi.org/10.7717/peerj.11075#supplemental-information.

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
