# Peer review of "Limited influence of marine sediment lyophilization on prokaryotic community structure assessed via amplicon sequencing: an example from environmentally contrasted sediment layers in Toulon harbor (France)"

_PeerJ, doi:10.7717/peerj.11075_

## Round 0.1 · original submission · Major Revisions

Despite the scientific interest of the paper, the reviewers and myself have found that the experimental design is weak. I strongly suggest that you clarify in the Materials and Methods section the replication scheme you used in your study, including biological and/or technical replicates. A simply drawing of your sampling scheme could be a helpful suggestion. Also, the statistical analyses related to these replicated samples should also be evident. Some of the reviewers comments include suggestions on how to rearrange your results towards this direction.

It is crucial for the fate of the paper to depict clearly that your results are based on truly replicated sample analysis.

Reviewer 1 ·

Basic reporting

The authors present an interesting manuscript whereby they investigated the effects of freeze drying (lyophilization) of samples for DNA based microbial community analysis. This is important because as the authors point out in their introduction, freeze drying is used to determine the metal burden per dry mass sediment, and it is important to know whether those same freeze dried samples could be used for DNA analysis. There are also other geochemical analyses that use freeze drying, so it is a very good idea to test the effects of lyophilization on the DNA protocols. The authors appear to have done a good job in producing their data, and use the up -to-date 16S primers with the Y ambiguity from Parada et al (modified EMP primers).

Experimental design

My main suggestion would be for the authors to do a qPCR of the 16S rRNA genes from the lyophilized and non-lyophilized samples, to check whether there is a quantitative difference in the amount of DNA extracted from both methods. If you do not have access to a qPCR, then at least you could present the total amount of DNA measured after the extraction? This is really important to know I think, in addition to the diversity effects shown in the sequencing data.

Validity of the findings

My main criticism is that looking at Figure 2 and Figure 3, it seems that in the 19-20 cm sample the lyophilized samples have a lower richness compared to the fresh samples. This should be stated in the text, and conclusions/discussion. If you just read the text you get the impression there is no difference , which is clearly not the case looking at Figure 2 in the deep sample. Also, in the 0-1 cm sample there is a huge difference between the lyophilization replicates. How do you explain that? Nevertheless, if you look at Figure 5 and Figure 6 there indeed does not appear to be a big difference between the treatments which is a good sign and indeed supports the authors conclusions.

Additional comments

General comment on the results: can you please add more information here? More details on sequencing and diversity statistics would be appreciated. Currently this is very sparse and general information that does not help the reader figure out what the really important trends are in the data. Also, please do not make paragraphs from a single sentence.

Discussion: please add to the discussion how well the DADA2 pipeline captures the real diversity. You are getting around 2000 ASVs which is realistic, but prior programs like UCLUST inflated diversity 10 to 100 times which was then corrected with USEARCH and confirmed using mock community controls (Edgar 2013). Might be worth commenting on this.

Additionally, it would be appreciated if in the discussion the authors could expand the comparison between the lyophilized and non-lyophilized samples. How many OTUs were found in one and not the other? Which groups were enriched in one and not the other? Was there a difference in the relative abundance of archaea, or bacteria, that are known to have different membrane structures that probably play a role in their being more or less easily lysed during the extraction procedure?


line 166: what do you mean "taxonomic affiliation precision", and how do you know what is "best"?

It is nice to see that microbial ecologists are starting to submit their work to PeerJ , and getting away from profit based journals (Nature, Frontiers, etc). This is a good sign for our field!

Reviewer 2 ·

Basic reporting

The authors aimed at comparing the microbial community composition in lyophilized and fresh marine harbor sediments from different depths. They hypothesized that lyophilization does not substantially affect cell lysis or DNA accessibility during extraction or PCR.
To this end from three sediment cores slices from three selected layers were split into halves and lyophilized or not. DNA extraction, PCR and MiSeq sequencing were performed using identical methods.
Based on rarefaction analysis, Alpha diversity, “taxonomic precision”, phylum/class level community structure and a PCoA of BC dissimilarity of samples the authors conclude that the sediment treatment does not appear to affect microbial community structure.

Experimental design

The scientific question addressed in this manuscript is interesting and relevant to many scientific sample sets. However, based on the presented data I do not concur with the authors conclusion that sediment treatment has low effects on MC structure. The sample size is to low, split samples should be compared (presented)in a pairwise manner and a statistical analysis should be performed. Figure 2c suggests a somewhat higher ASV richness in lyophilized sediments? In particular, the data depicted in Figure 5 question the conclusions made. MC structure depicted in stacked bars #2 (fresh) and #12 (lyophilized) seem not to have a good match to their corresponding samples. The authors should also show in more detail, which samples come from the same core/slice.
I also suggest performing a weighted UniFrac analysis to take phylogenetic distances between ASV into account.

Validity of the findings

In summary, the data do not justify the conclusion that intersample differences are the same or larger than between pretreatments.

Additional comments

Specific comments:
Ln61 and elsewhere: Please check formatting of in-text citations. Does it meet PeerJ requirements?
- Ln92: which instruments was used?
- Ln108 and elsewhere: check for spaces following numbers, 1000 ppm
- Ln113: three sediment slices
- Ln¬¬117 and elsewhere: PeerJ format? May read Fig.1 !?
- Ln117: 150 years old
- Ln136: trimmed
- Ln147: give details on used packages
- Ln160: 2,396
- Ln170 and elsewhere: check for italics
- Ln170: among 49 phyla
- Ln189: check for double spaces

Reviewer 3 ·

Basic reporting

The English is generally well written although the redaction of some sentences can be improved and have been indicated in the pdf file.

The introduction cover basically all the aspects required although some more details are needed and more clear text is necessary to convey some points. However, no basis is given on why the effect specifically of Hg can be potentially important.

Figures are generally of good quality although fig 4 can be improved.

Sequence data have been uploaded to a database.

Figures should be relevant to the content of the article, of sufficient resolution, and appropriately described and labelled.

The results are relevant to the hypothesis put forward but are not sufficient.

Experimental design

The subject is of interest and is still a matter of debate given the limited studies.

However, neither the questions are properly defined neither the design is adequate.

This is not a well designed looking into the effect of freeze drying vs fresh samples.

First, there are only two replicates which prevents any statistical analysis that will confirm the spurious observations made. Therefore, conclusions are a matter of personal perception. I actually interpreted the data in a different than the authors. In addition, taking into account that this is a potentially methodological paper, few details are given for the procedures. Finally, on this matter, authors used an additional pre-treatment for the fresh vs freeze dried sediment without explaining why and invalidating the comparison. I would also like to point out that what the authors label as fresh sediment is frozen sediment, it has not simple been freeze dried. This is very important as any freezing effect to the cells/sediment structure is similar.

Second, the authors intend to identify an effect of high Hg content in the sediment, however, they are unable to discern the effect of Hg from the effect of sediment depth. Unless they show that the OM content and quality and biogeochemical conditions are the same at 10 and 20 cm depth. Hg and depth are confounded.

Validity of the findings

Given the above, despite the potential interest of the study, given the lack of robustness in the design of the study, unless more replicate samples are used and a valid control site, conclusions are not verified.

More details are given in the attached pdf.

Annotated reviews are not available for download in order to protect the identity of reviewers who chose to remain anonymous.

---

## Round 0.2 · accepted · Accept

Thank you for your patience during the reviewing process. We hope to also receive your future works in PeerJ.

Reviewer 1 ·

Basic reporting

The authors have addressed all my comments.

Experimental design

The authors have addressed all my comments.

Validity of the findings

The authors have addressed all my comments.

Additional comments

The authors have addressed all my comments.

Reviewer 2 ·

Basic reporting

ok

Experimental design

ok

Validity of the findings

ok

Additional comments

The manuscript has been greatly improved and additional analysis has been peroformed (uniFrac, qPCR). I support publication.